# Workplace gender-based violence among female staff in public higher education institutions in eastern Ethiopia: Institution-based cross-sectional study

**Mowlid Abdi Ali**[1], **Agumasie Semahegn**[2,3,4]*, **Wondimye Ashenafi**[1,5], **Henok Legesse**[2]

**1** School of Public Health College of Health and Medical Sciences, Haramaya University, Harar, Ethiopia, **2** School of Nursing and Midwifery, College of Health and Medical Sciences, Haramaya University, Harar, Ethiopia, **3** Center for Innovative Drug Development and Therapeutic Trials for Africa (CDT-Africa), College of Health Sciences, Addis Ababa University, Addis Ababa, Ethiopia, **4** Department of Population, Family and Reproductive Health, School of Public Health, University of Ghana, Accra, Ghana, **5** Ethiopian Public Health Institute, National Data Management Center for Health, Addis Ababa, Ethiopia

* agumas04@gmail.com

**Data Availability Statement:** All data necessary to replicate this study findings publicly available

## Abstract

Despite the global and local efforts, gender-based violence at workplaces has remained a major public health challenge and pervasive human rights violation worldwide. Nevertheless, there is a paucity of research evidence on gender-based violence in higher educational institutions (HEIs). The main aim of this study was to assess workplace gender-based violence and its associated factors among female staff in public HEIs in eastern Ethiopia. An institution-based cross-sectional study design was conducted among female staff in public HEIs in eastern Ethiopia. Female staff (n = 391) were recruited using systematic sampling techniques from three HEIs. Data were collected by trained female data collectors using a structured pre-tested, self-administered questionnaire. Collected data were entered into EpiData and exported to SPSS for analysis. Descriptive and logistic regression statistical analysis were carried out to compute frequencies and odds ratio at 95% confidence interval (CI). The adjusted odds ratio (AOR) at 95% CI was used to declare a significant association. Workplace gender-based violence among female staff at HEIs was 63.1% (95% CI: 58–68%). Being within the age group of 18–34 years (AOR: 1.71, 95% CI: 1.02–2.85), being single (AOR: 2.24, 95% CI: 1.32–3.80), divorced (AOR: 2.27, 95% CI: 1.03–5.03), working the night shifts (AOR:5.73, 95% CI: 1.87–17.58), Being aware of the reporting procedures of violent incidents (AOR: 1.55, 95% CI: 1.01–2.49) and worried for being a victim of violence (AOR: 1.71, 95% CI: 1.02–2.86) were the factors associated with workplace gender-based violence against female staff in the public HEIs. Workplace gender-based violence among female staff working in the selected public HEIs was found to be unacceptably high. Awareness-raising campaigns against gender-based violence and reporting procedures in case of violent incidents, enforcing existing policies, orientation to employees, safeguarding the night shift female workers, and survivor support services should be implemented by key stakeholders.

without restriction at the time of publication (support information 4).

**Funding:** This study was financially supported by Haramaya University. MAA was a beneficiary of the research grant support, and all authors (MAA, AS, WA & HL) received salary from Haramaya University. The funders had no role in study design, data collection and analysis, decision to publish, or preparation of the manuscript. The interpretation of the finings is the responsibility of the researchers, particularly the lead author (MAA).

**Competing interests:** The authors have declared that no competing interests exist.

## Introduction

According to the United Nations (UN) definition in 1993, gender-based violence (GBV) is any act of violence that results in, or is likely to result in, physical, sexual, or mental harm or suffering to women, including threats of such acts, coercion, or arbitrary deprivation of liberty, whether occurring in public or in private life [1]. Globally, workplace GBV against women is the most pervasive but less recognized human rights abuse [1,2]. Many female staff have experienced different forms of workplace GBV, such as physical, sexual, and psychological violence, that disproportionately affected them at their workplace [2–6].

Workplace GBV is being recognized as a serious human rights abuse and has been a major public health challenge with substantial consequences on physical, mental, and reproductive wellbeing of female staff [7–12]. Globally, 38% of women were murdered by people around them [1], the extent of workplace GBV ranging from 10% to 69% [4], and caused a huge inequities between male and female staff. The prevalence of workplace GBV is ranges from 20% in the Western Pacific, 22% in Europe, and 33% in the WHO African region [1], to 67.7% in sub-Saharan African countries, including Ethiopia [9]. Workplace GBV in higher education institution (HEI) setting is an alarming and serious concern [13]. As a result of working in less safe environments and male dominated institutions, female staff are at high risk of workplace GBV [14]. However, the risk factors of workplace GBV are multi-dimensional, that interplay of social, economic, cultural, political, and religious dimensions either for being victim and/or perpetrator. In addition, existing evidence has revealed that senior male co-workers are the main perpetrators at the workplace due to power imbalances [15–17].

Existing evidence has reported that workplace violence is unacceptably high among female university or HEI staff in Ethiopia [18]. However, there is a paucity of research evidence on factors influencing workplace GBV, which could be the main reason for the poor implementation of workplace GBV prevention and control programs in Ethiopia [18]. Therefore, we conducted an institution-based study that aimed at assessing the extent of workplace GBV and its associated factors among female staff in the three public HEIs in eastern Ethiopia.

## Methods

### Ethics statement

The study protocol was reviewed and approved by Institutional Health Research Ethical Review Committee College of Health and Medical Science, Haramaya University (Reference number: IHRERC/176/2022). A formal letter was submitted to each university administration and permission was granted to conduct the data collection. The study was conducted in accordance with Helsinki's declaration. Informed verbal and written consent were obtained from each study participant in voluntary basis to be included in the study. Collected data were kept confidential anonymously through the de-identification of names and other personal identifiers from record/sheet, parents/guardians in the case of minor study participants and legally authorized representatives in case of illiterate participants.

### Study settings

A cross-sectional survey was conducted in three public HEIs in the eastern Ethiopia, namely: Haramaya, Dire Dawa, and Jigjiga Universities, from December 1 to 30, 2022. Haramaya University is located in Haramaya town, eastern Ethiopia, that is the second oldest public HEI in Ethiopia. It was established in 1954 as the 'Alemaya College of Agriculture' under Addis Ababa University until 1985. Currently, Haramaya University has massively been engaged in the expansion and diversification of academic programs for undergraduate, graduate, and

postgraduate training [19]. Both Dire Dawa University [20] and Jig-Jiga University [21] are established as second generation HEI or universities in 2007 in the eastern Ethiopia. Dire Dawa University has six colleges [20], similarly, Jig-Jiga University is found in Jig-Jiga City of the Somali Regional State, Ethiopia that is located, 635 kilometres from Addis Ababa, eastern Ethiopia. It has eight faculties that offer both undergraduate and postgraduate programs [21], and surrounded by a predominantly pastoral community in the Somali Region of Ethiopia who have also engaged in trade, cultivation of cash crops, and in transitive life [22].

## Study design and participants

An institutional-based cross-sectional study was conducted on the selected public universities (S1 Checklist). Female staff whose age was above 18 years and who had completed their probationary period of employment procedure (at least 6 months) in the selected HEIs were eligible for the study. Nevertheless, female staff who were on annual leave, maternity leave, and severe health conditions during the data collection period were excluded from the study.

## Sampling procedures

Sample size was determined by using a single population proportion formula ($n = (Z\alpha/2)^2 * P(1-P)/d^2$) considering parameters of workplace GBV prevalence (63.8%) in Nigeria (17), 95% significance level, 1% margin of error, and adding 10% potential non-response compensation. Finally, the sample size calculations yielded a total of 391 female staff. Then the calculated sample size was proportionally allocated to the number of female staff at selected public HEIs (**Fig 1**). A sampling frame was constructed for all female staff using the human resource records from selected public HEIs. A systematic sampling method was applied to recruit study participants. The first study participant from each university was selected using the lottery method, and then the next participants were selected using a pattern of every $k^{th}$ (k = 17) according to the respective sampling interval until the sample size was fulfilled from proportionally allocated study sites.

## Data collection procedure

Data were collected using a self-administered structured questionnaire that was adapted from existing literature, including the International Labor Office [2], a country-based survey questionnaire on workplace violence, and ending gender-based violence survey questionnaire on HEIs by UNICEF [23]. The questionnaire consisted of study participants' socio-demographic and workplace characteristics (12 items), institutional/contextual characteristics (9 items), GBV assessing questions for psychological violence (8 items), physical violence (4 items), and sexual violence and/or harassment (14 items), and responses towards the incidents (5 items). The questionnaire was first written in English and translated to local dialects (Amharic, Affaan-Oromo, and Somali). The backtranslation of questionnaire from the local language to English was also carried out by language experts to ensure consistency. Data were collected by 10 trained female nurses. Four data collectors for Haramaya University, three for Jig-Jiga University, and three for Dire Dawa University were assigned to facilitate the self-administered questionnaire distribution, clarification as deemed necessary, and re-collection of the filled questionnaire. Training was provided to data collectors on the study objectives, sampling, consent, data privacy, and checking for data clarity and completeness by the principal investigator. Then informed consent was taken by data collectors from the study participants and distributed the self-administered questionnaire in an envelope at their workplace in a confidential manner. The study participants were allowed to take enough time to fill out the questionnaire confidentially at their convenient time. The filled-out questionnaires were recollected by data collectors based on the timeline agreed upon with study participants in a voluntary basis.

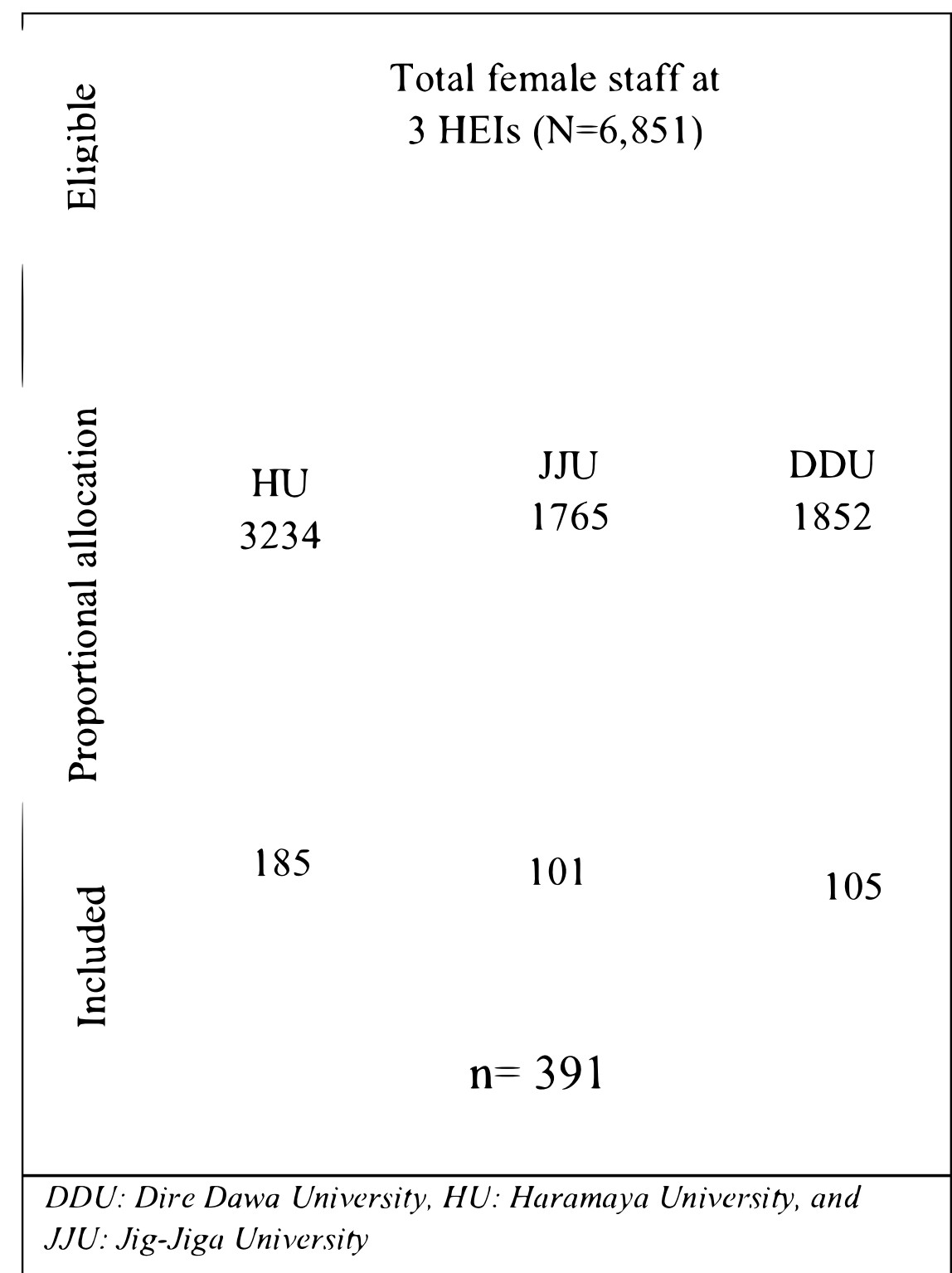

**Fig 1. Schematic presentation of sampling procedures.**

### Outcome variable measurements

Workplace GBV against female staff at HEIs was measured as a composite variable using the three very common types of workplaces GBV, namely physical, psychological, and sexual violence. So, in this study, workplace GBV was considered if a female staff whose age ranged from 15–49 years reported an experience of at least one form of violence (physical, psychological, and/or sexual violence) by co-workers, supervisors/bosses, visitors or strangers at their workplace in the past 12 months [10]. Psychological violence was assessed if a female staff has experienced of intentional use of power, threat of physical force, act of verbal abuse, insults and bullying or mobbing by co-workers, supervisors and/or visitors. Physical violence was assessed when female staff answered "yes" to one of the items: slapping, kicking, beating with any object, cutting, pushing, shoving, throwing, strangling, grabbing or pulling, and shooting against female staff. Sexual violence was considered if female staff reported any acts of violence at the workplace that were done on female staff by the intentional use of physical force or power, intimidation or threatening (making to fear) to have sex or to engage in acts of sex without the consent. It includes completed rape, attempted rape, and sexual harassment.

### Data processing and analysis

Collected data were entered into EpiData 3.1 and exported to IBM SPSS version 25 for cleaning and analysis. Descriptive statistical analyses were used to determine the proportion, mean and standard deviation for some of the explanatory variables and workplace GBV. Binary and multiple logistic regression analyses were performed to examine the association between the explanatory variables and workplace GBV. Explanatory variables with a p-value <0.25 in the binary logistic regression analysis were included in the multivariable logistic regression analysis. The Hosmer-Lemeshow test was used for goodness of fit for the final model's fitness. The adjusted odds ratio (AOR) at 95% CI was used to declare significance association between explanatory variables and workplace GBV.

## Results

### Socio-demographic characteristics of study participants

A total of 391 study participants were involved in the study. Of these, 385 of them completed the questionnaire with a response rate of 98.5% (Fig 1). The mean age of study participants was 32.39(±8.65) years. Nearly half of them were married (45.2%) and belonging to Orthodox Christianity (48.3%). Forty percent of the study participants had bachelor degree (Table 1).

### Workplace related characteristics

Out of 385, nearly half (47.5%) were from Haramaya University. The majority (90.4%) of female staff did not work the night shift, and 71.7% of them did not know to whom or where to report workplace GBV incidents. Slightly more than three-fourth (76.3%) of female staff were in the administrative department and nearly all of them (96.4%) had full-time permanent employment (Table 2).

### Workplace gender-based violence against female staff

The workplace GBV against female staff determined by considering forms of psychological, sexual and physical violence (Table 3). The prevalence of workplace GBV against female staff in HEIs was 63.1% (95% CI: 58–68%) (Fig 2).

**Table 1. Socio-demographic characteristics of the study participants in HEIs, eastern Ethiopia, 2022 (n = 385).**

| Variables | Categories | n | % |
|---|---|---|---|
| Age in years | 18–34 | 246 | 63.9 |
| | ≥35 | 139 | 36.1 |
| Marital status | Single | 146 | 37.9 |
| | Married | 174 | 45.2 |
| | Divorced | 43 | 11.2 |
| | Widowed | 22 | 5.2 |
| Religion | Muslim | 107 | 27.8 |
| | Orthodox | 186 | 48.3 |
| | Protestant | 78 | 20.3 |
| | Others* | 14 | 3.6 |
| Ethnicity | Oromo | 138 | 35.8 |
| | Amhara | 160 | 41.6 |
| | Somali | 61 | 15.8 |
| | Others** | 26 | 6.8 |
| Educational status | Secondary school | 35 | 9.1 |
| | Diploma level | 113 | 29.4 |
| | Bachelor degree | 154 | 40 |
| | Master degree and above | 83 | 21.5 |
| Salary level (Ethiopian Birr) | 1,100–2,798 | 117 | 30.4 |
| | 2,799–6,192 | 117 | 46.0 |
| | 6,193–10,150 | 71 | 18.4 |
| | ≥10,150 | 20 | 5.2 |
| Service years | 1–4 | 163 | 42.3 |
| | 5–9 | 146 | 37.3 |
| | ≥10 | 76 | 19.7 |
| Residence | On-campus | 70 | 18.2 |
| | Off-campus | 315 | 81.8 |

*Others include: Catholic, Waqefata, Jehovah's Witness, un-affiliated.
**Others include: Tigray, Gurage, Sidama, Harari.

## Perpetrators of the workplace GBV

One-third (34.8%) of workplace GBV against female staff were committed by their immediate supervisors/bosses, followed by 20.8% of them perpetrated by co-workers (Fig 3).

More than a quarter (28.1%) of female staff who had experienced workplace GBV were less productive. More than one-in-ten (11.7%) of them stayed off work after the incident. One-in-five (20.5%) of them claimed that they were emotionally exhausted after being victimized by violence at their workplace.

## Factors associated with workplace gender-based violence against female staff

In the binary logistic regressions, a total of twenty independent variables were included. Only eleven variables, namely; marital status, educational status, residence, work experiences, number of co-workers, age, staff category, night shift, reporting procedure, worry of violence, and institutional policy were found to be eligible for the final model with a p-value of <0.25 in the bivariate analysis. In the final multivariable logistic regression model, six explanatory variables

**Table 2. Institutional related characteristics of female staff working in public HEIs of eastern Ethiopia, 2022 (n = 385).**

| Variables | Categories | n | % |
|---|---|---|---|
| Institution | Haramaya University | 183 | 47.5 |
| | Jig-Jiga University | 98 | 25.5 |
| | Dire Dawa University | 104 | 27.0 |
| Staff category | Academic | 92 | 23.9 |
| | Administrative (non-academic) | 293 | 76.3 |
| Number of co-workers | 1–4 | 307 | 79.7 |
| | 5–9 | 45 | 11.7 |
| | ≥10 | 33 | 8.6 |
| Night shift | Yes | 37 | 9.6 |
| | No | 348 | 90.4 |
| Employment status | Contract (temporary) | 14 | 3.6 |
| | Permanent (full time) | 371 | 96.4 |
| Reporting procedures | Yes | 109 | 28.3 |
| | No | 276 | 71.7 |
| Institutional policy | Yes | 235 | 61.0 |
| | No | 150 | 39.0 |
| Worried of being vulnerable to violence | Not worried | 134 | 34.8 |
| | Little worried | 50 | 13.0 |
| | Moderately worried | 143 | 37.1 |
| | Much worried | 58 | 15.1 |

had a significant association with workplace GBV. Being single (AOR: 2.24, 95% CI: 1.32–3.80) and separated/divorced (AOR: 2.27, 95% CI: 1.03–5.03) were associated with increased odds of workplace GBV as compared with married female staff. Moreover, being within the age group of 18–34 years (AOR: 1.71, 95% CI: 1.02–2.85), working in the night shift

**Table 3. Types of workplace violence against female staff.**

| Type of violence | Items used for assessment | % |
|---|---|---|
| Psychological violence (57.7%) | Humiliating, diminishing, offensive or ridiculing | 42.3 |
| | Gave hostile looks, stares, sneers | 9.9 |
| | Gave abusive comment | 19.0 |
| | Made threaten comment | 16.6 |
| | Interrupted while talking or spoke over | 8.3 |
| | Unfairly rated performance lower than expected | 2.7 |
| | Ignored or did not want to talk with you | 0.8 |
| | Subjected to outburst anger | 0.5 |
| Physical violence (28.3%) | Threatened or hurt using physically force | 17.7 |
| | Pushed, shoved, slapped, grabbed, pulled | 4.7 |
| | Threw hard object to beat/kicked | 2.9 |
| | Tried to suffocate or strangle | 3.0 |
| Sexual violence (14.5%) | Attempt to export force to have sex | 5.7 |
| | Reported sexual favor an exchange | 3.6 |
| | Attempted to force for sexual intercourse | 2.9 |
| | Forced you for sexual intercourse | 1.0 |
| | Made to take part for sexual activity | 0.8 |
| | Show something sexual | 0.5 |

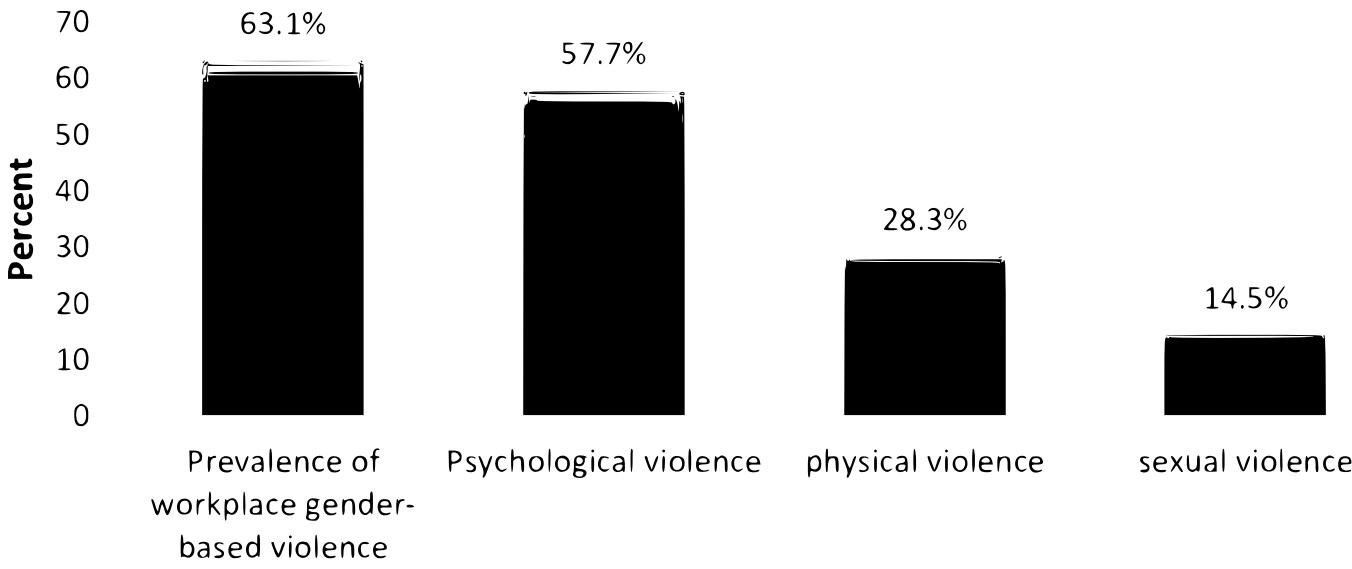

**Fig 2. Workplace GBV among female staff in public HEIs, eastern Ethiopia, 2022 (n = 385).**

(AOR:5.73, 95% CI: 1.87–17.58), being not aware of the existence of reporting procedures (AOR: 1.55, 95% CI: 1.01–2.49) and being worried about violence (AOR: 1.71, 95% CI: 1.02–2.86) were also associated with increased odds of workplace GBV as compared with their counterparts. On the other hand, the odds of experiencing workplace GBV decreased by 54%

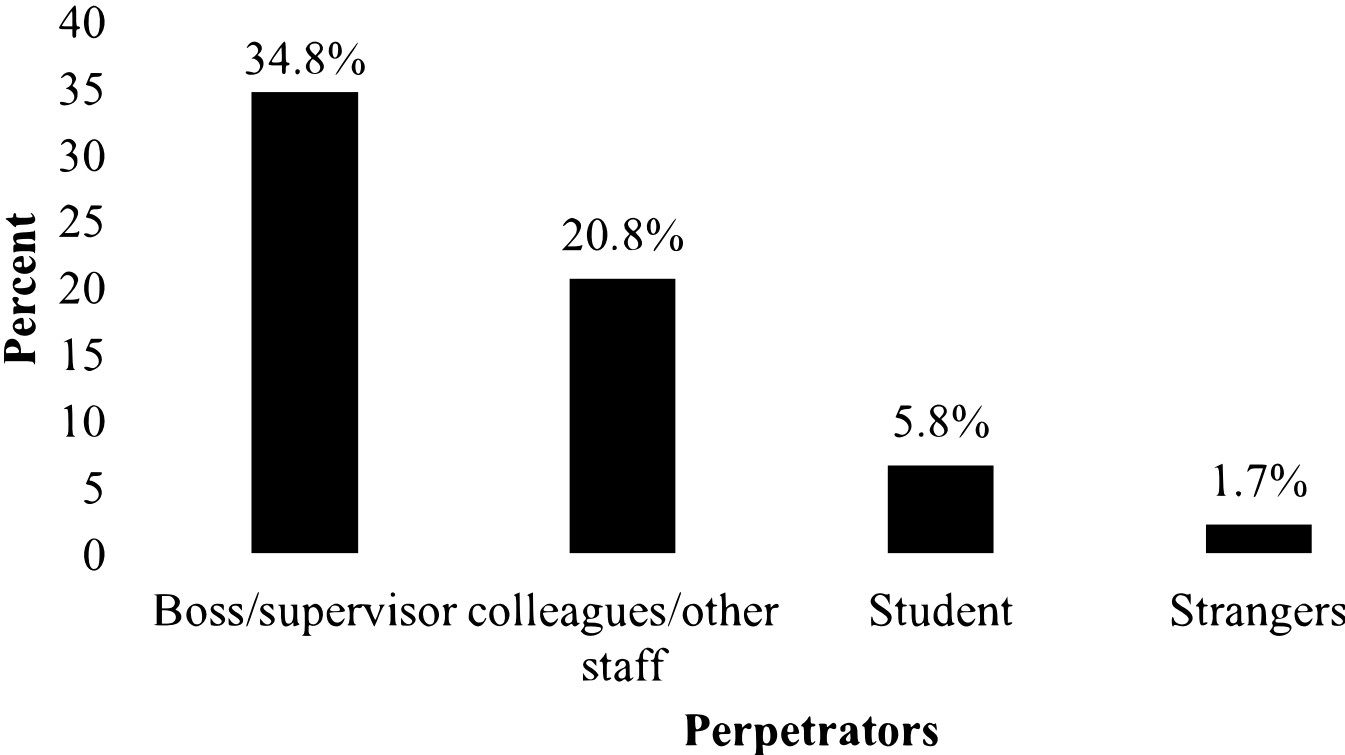

**Fig 3. Perpetrators of workplace GBV among female staff in public universities of eastern Ethiopia, 2022 (n = 385).**

**Table 4. Factors associated with GBV among female staff working in public HEIs in the eastern Ethiopia, 2022 (n = 385).**

| VARIABLE | CATEGORY | WORKPLACE GBV | | COR (95%CI) | AOR (95% CI) |
|---|---|---|---|---|---|
| | | Yes | No | | |
| MARITAL STATUS | Single | 110 | 36 | 2.66(1.6–4.44) | 2.24(1.32–3.80)* |
| | Divorced | 31 | 12 | 2.25(1.08–4.67) | 2.27(1.03–5.03) |
| | Widowed | 9 | 13 | 0.60(0.25–1.48) | 0.58(0.20–1.66)* |
| | Married | 93 | 81 | 1 | 1 |
| EDUCATIONAL STATUS | ≤Secondary | 23 | 12 | 0.61(0.26–1.44) | 0.81(0.31–2.14) |
| | Diploma | 76 | 37 | 0.65(0.34–0.24) | 0.90(0.44–1.85) |
| | BSc | 81 | 73 | 0.35(0.19–0.64) | 0.47(0.24–0.92)* |
| | ≥MSc | 63 | 20 | 1 | 1 |
| **RESIDENCE IN CAMPUS** | Inside | 49 | 21 | 1.45(0.83–2.55) | 1.48(0.78–2.85) |
| | Outside | 194 | 121 | 1 | 1 |
| WORK EXPERIENCE | 1–4 | 95 | 68 | 1.02(0.59–1.76) | 0.72(0.39–1.33) |
| | 5–9 | 104 | 42 | 1.80(1.01–3.22) | 1.27(0.67–2.41) |
| | ≥10 | 44 | 32 | 1 | 1 |
| **CO-WORKERS'** | 1–4 | 183 | 124 | 0.40(0.17–0.94) | 0.61(0.24–1.58) |
| | 5–9 | 34 | 11 | 0.83(0.28–2.44) | 1.28(0.39–4.12) |
| | ≥10 | 26 | 7 | 1 | 1 |
| AGE | 18–34 | 171 | 75 | 2.12(1.38–3.26) | 1.71(1.02–2.85)* |
| | ≥35 | 72 | 67 | 1 | 1 |
| **STAFF CATEGORY** | Academic Administrative | 63 180 | 29 113 | 1.36(0.83–2.25) 1 | 1.33(0.74–2.41) 1 |
| **NIGHT SHIFT** | Yes | 33 | 4 | 5.42(1.88–15.64) | 5.73(1.87–17.58 |
| | No | 210 | 138 | 1 | 1 |
| KNEW GBV REPORTING PROCEDURE | Yes | 126 | 88 | 1 | 1 |
| | No | 117 | 54 | 1.51(0.99–2.31) | 1.55(1.01–2.49)* |
| **WORRIED FOR BEING RISK TO VIOLENCE** | Worried | 92 | 42 | 1.45(0.93–2.26) | 1.71(1.02–2.86)* |
| | Not worried | 151 | 100 | 1 | 1 |
| **INSTITUTIONAL POLICY AGAINST GBV** | Yes | 142 | 93 | 0.74(0.48–1.14) | 0.74 (0.49–1.22) |
| | No | 101 | 49 | 1 | 1 |

*Significantly associated at a p-value of <0.05.

among female staff with BSc degree compared to female staff with Master Degree and above (AOR:0.46, 95% CI:0.24–0.90) (Table 4).

## Discussion

This study determined the prevalence of workplace GBV and associated factors among female staff in the public HEIs. Nearly two-thirds (63.1%, 95%CI: 58–68%) of female staff at HEIs experienced workplace GBVs during the last 12 months. The present study finding is similar to the prevalence of workplace GBV against female staff (63.8%) in Nigeria [24] and 63.2% in Iran [25]. On the other hand, this finding is higher than the findings from studies conducted in Nepal (49.5%) [6], 44.7% in Turkey [8], 58.5% in Rwanda [10], and 35.1% in different parts of Ethiopia [26]. Nevertheless, the findings of this study are slightly lower than the findings from studies conducted in China (77.5%) [27] and 70% in Kenya [28]. This discrepancy is might be due to cultural, socio-economic differences and level of enforcement of existing regulations, strategies, and policies to respond to gender-based violence in those countries.

In the present study psychological violence is the most common form of workplace GBV against female staff (57.7%) which is similar to the findings from a couple of studies conducted in Nigeria [29] and Gambia [30]. Furthermore, in the present study, sexual violence was 14.5%, which is similar to a study conducted in Nepal [6]. Nevertheless, sexual violence of this findings is much lower than in other studies conducted in other parts of Ethiopia, which have reported that 49.1% in Bahir Dar [31], 39.2% in Dessie [32], 56.4% in Harar [33]. The findings of this study showed that sexual violence is high in younger female staff compared to older age, which is similar with study conducted in Ethiopia [34]. Similarly, the present study revealed that 28.3% of female staff were subjected to physically violence which is consistent with studies conducted in Zimbabwe [35] and Ethiopia. where nearly one-third of women experienced at least one form of GBV [33,36], and United Kingdom [14].

According to the findings of the current survey, having bachelor degree is significantly associated with less experience of workplace GBV, which is consistent with a finding in Taiwan [37] and in Nigeria among female university staff [24]. It might be due to the protective effect of education on women against disrespectful or inappropriate behavior at the workplace [38,39]. The findings of this study revealed that female staff whose age were between 18–34 years experienced more GBV at workplace, which is similar with the study conducted in another part of Ethiopia [3]. Furthermore, the present study reveals that as age of female staff increased, the risks of workplace GBV decreased among female staff at HEIs in eastern Ethiopia. It might be obvious that elders are relatively respected in many cultures in Ethiopia and relatively less susceptible for GBV at workplace compared to young female staff, and may elders less exhibit their experience of GBV at workplace. This finding is consistent with a study conducted in Nigeria [24], as well as study conducted in China [40]. In the present study, participants working night shifts were 5.5 times more likely to experience workplace GBV, which is similar to the study conducted in Egypt [41] and other University in Ethiopia [42], and also public hospitals in the eastern Ethiopia [36].

The odds of workplace GBV among female staff who were not aware of reporting procedures were higher than among female staff who had awareness of reporting procedures within the HEIs. These factors were also reported in a study conducted in Saudi Arabia [43] and in Ethiopia [36]. The findings of this study demonstrated that being worried about violence at workplace was substantially associated with female staff experiencing workplace GBV. The identified factors from the present study were similar to those found in a study conducted in Brazil [44]. However, all HEIs abide by the laws and existing legal frameworks of the country, such as the Constitution of the Federal Democratic Republic of Ethiopia [45] that provides fundamental liberties, promotes gender equality, and safeguards women's human rights. The Criminal Code of Ethiopia under Proclamation No. 414/2004 [46] guarantees equality before the law (Art. 4) and criminalizes any injury and suffering caused to women (Art. 561) [47], and equal rights in the management of the government of Ethiopia launched a gender mainstreaming program in different sectors with an implementation manual to enforce existing policies [48]. Furthermore, the Ethiopian Ministry of Health has developed standard operating procedures to respond to and prevent GBV [49], and other gender equality related legislations and recommendations [49–51]. By principle, all HEIs in Ethiopia have zero tolerance policy towards workplace GBV.

## Strengths and weaknesses of the study

The study used a validated and comprehensive tool that encompasses the magnitude, nature, and effect of workplace GBV in a relatively broader fashion. Although this study assessed the extent of gender-based violence, gender disparity was not assessed in terms of education

opportunities, position, or leadership. In addition, there is the possibility of response biases since research evidence suggests that female staff tend not to report workplace GBV experiences for fear of stigma or retribution. This study might also be prone to recall-bias. The causal relationship between explanatory variables and workplace GBV may be limited due to temporality challenges in the cross-sectional study design.

## Conclusions

The prevalence of workplace GBV among female staff at higher education institutions in eastern Ethiopia was found to be high. Female staff experience of forms of GBV in the workplace was significantly associated with their age, lower academic qualification, marital status, reporting procedures, and work experience of 4–9 years. A significant proportion of female staff at HEIs were mainly perpetrated by their work supervisors and/or bosses. Therefore, it is recommended to establish a workplace GBV mitigation committee, create awareness on how to respond against workplace GBV and seek support, track the incidents, provide guidelines, and make sure all corrective measures are taken and communicated properly.

## Supporting information

**S1 Checklist. STROBE statement—Checklist of items that should be included in reports of observational studies.**
(DOCX)

**S1 Data.**
(DTA)

## Author Contributions

**Conceptualization:** Mowlid Abdi Ali, Agumasie Semahegn, Wondimye Ashenafi, Henok Legesse.

**Data curation:** Mowlid Abdi Ali, Agumasie Semahegn, Wondimye Ashenafi, Henok Legesse.

**Formal analysis:** Mowlid Abdi Ali, Henok Legesse.

**Investigation:** Mowlid Abdi Ali, Agumasie Semahegn.

**Methodology:** Mowlid Abdi Ali, Agumasie Semahegn, Wondimye Ashenafi, Henok Legesse.

**Project administration:** Mowlid Abdi Ali.

**Resources:** Mowlid Abdi Ali, Agumasie Semahegn.

**Software:** Mowlid Abdi Ali, Agumasie Semahegn, Henok Legesse.

**Supervision:** Mowlid Abdi Ali, Agumasie Semahegn, Henok Legesse.

**Validation:** Agumasie Semahegn, Wondimye Ashenafi, Henok Legesse.

**Visualization:** Mowlid Abdi Ali, Agumasie Semahegn.

**Writing – original draft:** Mowlid Abdi Ali, Agumasie Semahegn, Wondimye Ashenafi, Henok Legesse.

**Writing – review & editing:** Mowlid Abdi Ali, Agumasie Semahegn, Wondimye Ashenafi, Henok Legesse.

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
