## [Decision Letter · Decision Letter 0]

16 Aug 2023

PGPH-D-23-01225

Workplace gender-based violence among female staff in public higher education institutions, eastern Ethiopia: Institution-based cross-sectional study

Dear Dr. Semahegn,

Thank you for submitting your manuscript to PLOS Global Public Health. After careful consideration, we feel that it has merit but does not fully meet PLOS Global Public Health’s publication criteria as it currently stands. Therefore, we invite you to submit a revised version of the manuscript that addresses the points raised during the review process.

We look forward to receiving your revised manuscript.

Kind regards,

Jianhong Zhou

Staff Editor

Journal Requirements:

1. We noticed you have some minor occurrence of overlapping text with the following previous publication(s), which needs to be addressed:

- https://www.yada.org.tr/

- https://doi.org/10.1186/s12905-022-02001-8

In your revision ensure you cite all your sources (including your own works), and quote or rephrase any duplicated text outside the methods section. Further consideration is dependent on these concerns being addressed.

Additional Editor Comments (if provided):

Reviewers' comments:

Reviewer's Responses to Questions

**Comments to the Author**

1. Does this manuscript meet PLOS Global Public Health’s publication criteria? Is the manuscript technically sound, and do the data support the conclusions? The manuscript must describe methodologically and ethically rigorous research with conclusions that are appropriately drawn based on the data presented.

Reviewer #1: Yes

Reviewer #2: Partly

2. Has the statistical analysis been performed appropriately and rigorously?

Reviewer #1: Yes

Reviewer #2: Yes

3. Have the authors made all data underlying the findings in their manuscript fully available (please refer to the Data Availability Statement at the start of the manuscript PDF file)?

Reviewer #1: Yes

Reviewer #2: Yes

4. Is the manuscript presented in an intelligible fashion and written in standard English?

Reviewer #1: Yes

Reviewer #2: Yes

5. Review Comments to the Author

Reviewer #1: The manuscript is very well written and touched a very important & sensitive issue. For more and clear information I, have attached the file. The manuscript need minor revision. How the participants were contacted and how questionnaire were distributed and collected back should be discussed in detail.

Reviewer #2: On the grounds that much of the literature on GBV is from a western perspective, this paper should be considered for publication. Yet it needs major revision, in my opinion.

I am attaching a marked up copy for authors.

The following points are made

1. The language needs attention – I have highlighted in green the spelling errors, sentence construction errors, incorrect word usage or tenses that I spotted as I reviewed. Proof reading will be required on the final submission

2. The reporting of other studies, and indeed the authors own findings need to be reported in more detail and with the degree of precision required for an academic journal. Simple comparisons of percentages are only valid if the authors can clarify that the same measurement instruments and cut off points were employed across studies. Otherwise the comparisons are of limited utility. This needs to addressed.

3. There is no contextual information on the cultural backdrop to the study. Why are these rates so high? What are other HEIs doing to address this problem? Can the Ethiopian Universities use similar approaches? The data as presented is of interest but the discussion lacks depth and consideration. A more probing and insightful analysis and discussion is required for publication. There is virtually no reference to the large international literature on this topic

4. There is an implicit assumption that gender based violence is violence perpetrated by men, upon women. This is a limited view and persons who do not conform to a stereotypical sex role or gender identity may be a victim of GBV. Authors need to demonstrate knowledge of the nuances and complexities of this area.

6. PLOS authors have the option to publish the peer review history of their article (what does this mean?). If published, this will include your full peer review and any attached files.

**Do you want your identity to be public for this peer review?** For information about this choice, including consent withdrawal, please see our Privacy Policy.

Reviewer #1: **Yes: **Anuradha Nadda

Reviewer #2: No

---

## [Decision Letter · Decision Letter 1]

29 Apr 2024

PGPH-D-23-01225R1

Workplace gender-based violence among female staff in public higher education institutions, eastern Ethiopia: Institution-based cross-sectional study

Dear Dr. Semahegn,

Thank you for submitting your manuscript to PLOS Global Public Health. After careful consideration, we feel that it has merit but does not fully meet PLOS Global Public Health’s publication criteria as it currently stands. Therefore, we invite you to submit a revised version of the manuscript that addresses the points raised during the review process.

We look forward to receiving your revised manuscript.

Kind regards,

Lu Gram, Ph.D.

Academic Editor

Journal Requirements:

2. Please provide separate figure files in .tif or .eps format only and remove any figures embedded in your manuscript file. Please also ensure all files are under our size limit of 10MB.

Additional Editor Comments (if provided):

Thank you for your submission, I have read through the revised version and reviewer comments and am satisfied that most of the important substantive points have been addressed (you can take Reviewer 2's comments somewhat lightly, as they mainly about not looking at violence against men, which is outside the scope of the article). However, one of PLOS's publication criteria is clear, correct, and intelligible English. We suggest you have a fluent, preferably native, English-language speaker thoroughly copy-edit your manuscript for language usage, spelling, and grammar. If you do not know anyone who can do this, you may wish to consider employing a professional scientific editing service. Whilst you may use any professional scientific editing service of your choice, PLOS has partnered with both American Journal Experts (AJE) and Editage to provide discounted services to PLOS authors.

Reviewers' comments:

Reviewer's Responses to Questions

**Comments to the Author**

1. If the authors have adequately addressed your comments raised in a previous round of review and you feel that this manuscript is now acceptable for publication, you may indicate that here to bypass the “Comments to the Author” section, enter your conflict of interest statement in the “Confidential to Editor” section, and submit your "Accept" recommendation.

Reviewer #1: All comments have been addressed

Reviewer #3: (No Response)

2. Does this manuscript meet PLOS Global Public Health’s publication criteria? Is the manuscript technically sound, and do the data support the conclusions? The manuscript must describe methodologically and ethically rigorous research with conclusions that are appropriately drawn based on the data presented.

Reviewer #1: Yes

Reviewer #3: No

3. Has the statistical analysis been performed appropriately and rigorously?

Reviewer #1: Yes

Reviewer #3: No

4. Have the authors made all data underlying the findings in their manuscript fully available (please refer to the Data Availability Statement at the start of the manuscript PDF file)?

Reviewer #1: Yes

Reviewer #3: (No Response)

5. Is the manuscript presented in an intelligible fashion and written in standard English?

Reviewer #1: Yes

Reviewer #3: Yes

6. Review Comments to the Author

Reviewer #1: Author has done all the changes as suggested in last revision. Manuscript has touched a very important and sensitive area.

Reviewer #3: Thank you for inviting me to review this interesting research area: gender

I have the following concerns and feedback on the work:

1. Definition and approach: though I agree that females are disproportionately affected by violelences, I dont agree that concept gender is not all about females. There should be points that put more males at risk than females sometimes. Studying females alone is possible but not the only way. I would encourage the authors to clearly define gender violence and appreciate other rrlavant terms such as- gender vs sex vs sex orientation, sexual violence, pyschological violence, physical violences too. How do you see studying gender(social definition) but sampling female participants ( biological definition). In the UN definition you considered the word "women". Who are women in your study? Biologically females? Does it include those who consider themselves as women though biologically male at birth? Are there anyone who consider themselves as women while being male at birth in the study setting? How did you know? Do you have preminary assessment about this? Or have you checked how "females" included in this study consider themselves? What if they consider themselves as men? Overall, do not consider definitoon of gender superficially because it leads to superficial responses or findings. Please improve yiut backgrouns by clearly out those points by writing in the manuscript.

2. Methods: measurement is one of the methodological issues not well addressed in this study. I found that this study has almost no new finding. It is okay to repeat studies but efforts should be there to add new knowledge to literatute. Measurement issues of this study include: 1) how this study understands gender and how it measured key aspects of gender is a big limitation. Gender has sociocultural, organizational, systematic or structural factors or drivers. Did you assess empowerment (education, economy, leadership roles), decision making, etc.? That is not reflected. Both qualitativr and wuantitativr methods was required to shade light on those things-how they shaped "gender violences". 2) the meaurement of sexual violence is superficial. How did you measure and report sexual harassment? Please report such things as reported ones because harasment is too subjective to define. That may be the reason why the estimate of violence is as high as 63% in your study.

Sexual violence has 3 aspects in this study: rape, attempted rape, and harasment. Better to report these separately for the readers to know the estimates for each. 3) pyschological violence is more blurred in this study. Please define, measure, and report them clearly. Please provide proper descriptions for each items included to measure it. This should also be done for sexual and physical aspects too. Now it is over merged. Please give % of each items in the description section. For associated factor analysis you may used merged or latent variable. 4) why qualitative data are missed study given its great relevance?

3: Discussion and conclusion: properly dicuss gender violence intrms of sociocultural and systemic factors. I see a great deal of limitations in this accord that should be clearly and honestly indicated in this manuscript. I hope reanalysis of gender violence and its specific aspects will provide different estimate.

4: general question: have you reported this huge esrimate of viplence to your institution leaderships and collected their remarkks? It can also be considered as qualitative data and even reported in this study.

7. PLOS authors have the option to publish the peer review history of their article (what does this mean?). If published, this will include your full peer review and any attached files.

**Do you want your identity to be public for this peer review?** For information about this choice, including consent withdrawal, please see our Privacy Policy.

Reviewer #1: No

Reviewer #3: No

---

## [Editor Report · Decision Letter 2]

14 May 2024

PGPH-D-23-01225R2

Workplace gender-based violence among female staff in public higher education institutions, eastern Ethiopia: Institution-based cross-sectional study

Dear Dr. Semahegn,

Thank you for submitting your manuscript to PLOS Global Public Health. After careful consideration, we feel that it has merit but does not fully meet PLOS Global Public Health’s publication criteria as it currently stands. Therefore, we invite you to submit a revised version of the manuscript.

In particular, I've reviewed the manuscript and found that it still suffers from many grammatical mistakes and a lack of clarity in terms of its English language. The changes made since the last version do not fix the many issues with grammar and presentation. I have therefore no choice but to send the manuscript back to you for another round of English language editing. I would suggest that you get hold of a fluent, preferably native-level English speaker to copy-edit your manuscript. If you do not know of one, you can hire professional services (see below). However you get it done, please make sure to look over the language issues properly this time.

If you do not know anyone who can help proofread your paper, you may wish to consider employing a professional scientific editing service. Whilst you may use any professional scientific editing service of your choice, PLOS has partnered with both American Journal Experts (AJE) and Editage to provide discounted services to PLOS authors. Both organizations have experience helping authors meet PLOS guidelines and can provide language editing, translation, manuscript formatting, and figure formatting to ensure your manuscript meets our submission guidelines. To take advantage of the partnership with AJE, visit the AJE website (https://www.aje.com/go/plos/) for a 15% discount off AJE services. To take advantage of the partnership with Editage, visit the Editage website (www.editage.com) and enter referral code PLOSEDIT for a 15% discount off Editage services.

We look forward to receiving your revised manuscript.

Kind regards,

Lu Gram, Ph.D.

Academic Editor

Journal Requirements:

1. Please review your reference list to ensure that it is complete and correct. If you have cited papers that have been retracted, please include the rationale for doing so in the manuscript text, or remove these references and replace them with relevant current references. Any changes to the reference list should be mentioned in the rebuttal letter that accompanies your revised manuscript. If you need to cite a retracted article, indicate the article’s retracted status in the References list and also include a citation and full reference for the retraction notice

If you did not receive any funding for this study, please simply state: “The authors received no specific funding for this work.

3. Please provide separate figure files in .tif or .eps format only and remove any figures embedded in your manuscript file. Please also ensure all files are under our size limit of 10MB.

4. We have noticed that you have uploaded Supporting Information files, but you have not included a list of legends. Please add a full list of legends for your Supporting Information files after the references list.

---

## [Editor Report · Decision Letter 3]

27 May 2024

PGPH-D-23-01225R3

Workplace gender-based violence among female staff in public higher education institutions in eastern Ethiopia: Institution-based cross-sectional study

Dear Dr. Semahegn,

Thank you for submitting your manuscript to PLOS Global Public Health. After careful consideration, we feel that it has merit but does not fully meet PLOS Global Public Health’s publication criteria as it currently stands.

Although the revisions have improved the English of the manuscript somewhat, it still suffers from multiple obvious and glaring spelling and grammatical mistakes - such as "lobal" instead of "global" in the very first line of the abstract. As a result, I'm sending the manuscript back to you for another round of English language editing. I would suggest that you get hold of a fluent, preferably native-level English speaker to copy-edit your manuscript.

If you do not know anyone who can help proofread your paper, you may wish to consider employing a professional scientific editing service. Whilst you may use any professional scientific editing service of your choice, PLOS has partnered with both American Journal Experts (AJE) and Editage to provide discounted services to PLOS authors. Both organizations have experience helping authors meet PLOS guidelines and can provide language editing, translation, manuscript formatting, and figure formatting to ensure your manuscript meets our submission guidelines. To take advantage of the partnership with AJE, visit the AJE website (https://www.aje.com/go/plos/) for a 15% discount off AJE services. To take advantage of the partnership with Editage, visit the Editage website (www.editage.com) and enter referral code PLOSEDIT for a 15% discount off Editage services.

Therefore, we invite you to submit a revised version of the manuscript.

Please ensure that your decision is justified on PLOS Global Public Health’s publication criteria and not, for example, on novelty or perceived impact.

We look forward to receiving your revised manuscript.

Kind regards,

Lu Gram, Ph.D.

Academic Editor

Journal Requirements:

1. Please update your online Competing Interests statement. If you have no competing interests to declare, please state: “The authors have declared that no competing interests exist.”

2. Please provide separate figure files in .tif or .eps format only and remove any figures embedded in your manuscript file. Please also ensure that all files are under our size limit of 10MB. You may leave the figure captions or legends in the manuscript.
---

## [Editor Report · Decision Letter 4]

24 Jun 2024

Workplace gender-based violence among female staff in public higher education institutions in eastern Ethiopia: institution-based cross-sectional study

PGPH-D-23-01225R4

Dear Dr. Semahegn,

We are pleased to inform you that your manuscript 'Workplace gender-based violence among female staff in public higher education institutions in eastern Ethiopia: institution-based cross-sectional study' has been provisionally accepted for publication in PLOS Global Public Health.

Best regards,

Lu Gram, Ph.D.

Academic Editor